# CLASSIFY OR SELECT: NEURAL ARCHITECTURES FOR EXTRACTIVE DOCUMENT SUMMARIZATION

**Ramesh Nallapati, Bowen Zhou**
IBM Watson
Yorktown Heights, NY 10598 USA
{nallapati,zhou}@us.ibm.com

**Mingbo Ma**
Oregon State University
Kelley Engineering Center, Corvallis, OR, 97331
mam@oregonstate.edu

## ABSTRACT

We present two novel and contrasting Recurrent Neural Network (RNN) based architectures for extractive summarization of documents. The *Classifier* based architecture sequentially accepts or rejects each sentence in the original document order for its membership in the final summary. The *Selector* architecture, on the other hand, is free to pick one sentence at a time in any arbitrary order to piece together the summary.

Our models under both architectures jointly capture the notions of salience and redundancy of sentences. In addition, these models have the advantage of being very interpretable, since they allow visualization of their predictions broken up by abstract features such as information content, salience and redundancy.

We show that our models reach or outperform state-of-the-art supervised models on two different corpora. We also recommend the conditions under which one architecture is superior to the other based on experimental evidence.

## 1 INTRODUCTION

Document summarization is an important problem that has many applications in information retrieval and natural language understanding. Summarization techniques are mainly classified into two categories: extractive and abstractive. Extractive methods aim to select salient snippets, sentences or passages from documents, while abstractive summarization techniques aim to concisely paraphrase the information content in the documents.

A vast majority of the literature on document summarization is devoted to extractive summarization. Traditional methods for extractive summarization can be broadly classified into greedy approaches (*e.g.*, Carbonell & Goldstein (1998)), graph based approaches (*e.g.*, Radev & Erkan (2004)) and constraint optimization based approaches (*e.g.*, McDonald (2007)).

Recently, neural network based approaches have become popular for extractive summarization. For example, Kageback et al. (2014) employed the recursive autoencoder (Socher et al. (2011)) to summarize documents, producing best performance on the Opinosis dataset (Ganesan et al. (2010)). Yin & Pei (2015) applied Convolutional Neural Networks (CNN) to project sentences to continuous vector space and then select sentences by minimizing the cost based on their 'prestige' and 'diverseness', on the task of multi-document extractive summarization. Another related work is that of Cao et al. (2016), who address the problem of query-focused multi-document summarization using query-attention-weighted CNNs.

Recently, with the emergence of strong generative neural models for text Bahdanau et al. (2014), abstractive techniques are also becoming increasingly popular (Rush et al. (2015), Nallapati et al. (2016b) and Nallapati et al. (2016a)). Despite the emergence of abstractive techniques, extractive techniques are still attractive as they are less complex, less expensive, and generate grammatically

and semantically correct summaries most of the time. In a very recent work, Cheng & Lapata (2016) proposed an attentional encoder-decoder for extractive single-document summarization and trained it on Daily Mail corpus, a large news data set, achieving state-of-the-art performance. Like Cheng & Lapata (2016), our work also focuses only on *sentential* extractive summarization of single documents using neural networks.

## 2 TWO ARCHITECTURES

Our architectures are motivated by two intuitive strategies that humans tend to adopt when they are tasked with extracting salient sentences in a document. The first strategy, which we call *Classify*, involves reading the whole document once to understand its contents, and then traversing through the sentences in the original document order and deciding whether or not each sentence belongs to the summary. The other strategy that we call *Select* involves memorizing the whole document once as before, and then picking sentences that should belong to the summary one at a time, in any order of one's choosing. Qualitatively, the latter strategy appears to be a better one since it allows us to make globally optimal decisions at each step. While it may be harder for humans to follow this strategy since we are forgetful by nature, one may expect that the Select strategy could deliver an advantage for the machines, since 'forgetfulness' is not a real 'concern' for them. In this work, we will explore both the strategies empirically and make a recommendation on which strategy is optimal under what conditions.

Broadly, our *Classify* architecture involves an RNN based sequence classification model that sequentially classifies each sentence into 0/1 binary labels, while the *Select* architecture involves a generative model that sequentially generates the indices of the sentences that should belong to the summary. We will first discuss the components shared by both the architectures and then we will present each architecture separately.

**Shared Building Blocks**: Both architectures begin with word-level bidirectional Gated Recurrent Unit (GRU) based RNNs (Chung et al. (2014)) run independently over each sentence in the document, where each time-step of the RNN corresponds to a word index in the sentence. The average pooling of the concatenated hidden states of this bidirectional RNN is then used as an input to another bidirectional RNN whose time steps correspond to sentence indices in the document. The concatenated hidden states 'h' from the forward and backward layers of this second layer of bidirectional RNN at each time step are used as corresponding sentence representations. We also use the average pooling of the sentence representations as the document representation 'd'. Both architectures also maintain a dynamic summary representation 's' whose estimation is architecture dependent. Models under each architecture compute a score for each sentence towards its summary membership. Motivated by the need to build humanly interpretable models, we compute this score by explicitly modeling abstract features such as salience, novelty and information content as shown below:

$$
\begin{aligned}
\text{score}(\mathbf{h}_j, \mathbf{s}_j, \mathbf{d}, \mathbf{p}_j) = \; & w_c \sigma(\mathbf{W}_c^T \mathbf{h}_j) && \texttt{\#(content richness)} \\
& + w_s \sigma(\cos(\mathbf{h}_j, \mathbf{d})) && \texttt{\#(salience w.r.t. document)} \\
& + w_p \sigma(\mathbf{W}_p^T \mathbf{p}_j) && \texttt{\#(positional importance)} \\
& - w_r \sigma(\cos(\mathbf{h}_j, \mathbf{s}_j)) && \texttt{\#(redundancy w.r.t. summary)} \\
& + b, && \texttt{\#(bias term)}
\end{aligned}
\tag{1}
$$

where $j$ is the index of the sentence in the document, $\mathbf{p}_j$ is the positional embedding of the sentence computed by concatenation of embeddings corresponding to forward and backward position indices of the sentence in the document; $\cos(\mathbf{a}, \mathbf{b})$ is the standard cosine similarity between the two vectors $\mathbf{a}$ and $\mathbf{b}$; $\mathbf{W}_c$ and $\mathbf{W}_p$ are parameter vectors to model content richness and positional importance of sentences respectively; and $w_c, w_s, w_p$ and $w_r$ are scalar weights to model relative importance of various abstract features, and are learned automatically. In the equation above, the abstract feature that each term represents is printed against the term in comments. In other words, assuming the importance weights are positive, in order for a sentence to score high for summary membership, it needs to be highly salient, content rich and occupy important positions in the document, while being least redundant with respect to the summary generated till that point. Note that our formulation of the scoring function simultaneously captures both salience of the sentence $\mathbf{h}_j$ with respect to the document $\mathbf{d}$ as well as its redundancy with respect to the current summary representation $\mathbf{s}_j$. In

the next subsection, we will describe the estimation of dynamic summary representation $\mathbf{s}_j$ and the formulation of the cost function for training in each architecture. We will also present shallow and deep models under each architecture.

## 2.1 CLASSIFIER ARCHITECTURE

In this architecture, we sequentially visit each sentence in the original document order and binary-classify the sentence in terms of whether it belongs to the summary. The probability of the sentence belonging to the summary, $P(y_j = 1)$ is given as follows:

$$P(y_j = 1|\mathbf{h}_j, \mathbf{s}_j, \mathbf{d}, \mathbf{p}_j) = \sigma(\texttt{score}(\mathbf{h}_j, \mathbf{s}_j, \mathbf{d}, \mathbf{p}_j)) \quad (2)$$

The objective function to minimize at training is the negative log-likelihood of the training data labels:

$$\ell(\mathbf{W}, \mathbf{w}, \mathbf{b}) = -\sum_{d=1}^{N}\sum_{j=1}^{N_d}(y_j^d \log P(y_j^d = 1|\mathbf{h}_j^d, \mathbf{s}_j^d, \mathbf{d}_d) + (1 - y_j^d)\log(1 - P(y_j^d = 1|\mathbf{h}_j^d, \mathbf{s}_j^d, \mathbf{d}_d))$$

where $N$ is the size of the training corpus and $N_d$ is the number of sentences in the document $d$. Now the only detail that remains is how the dynamic summary representation $\mathbf{s}_j$ is estimated. This is where the shallow and deep models under this architecture differ, and we describe them below.

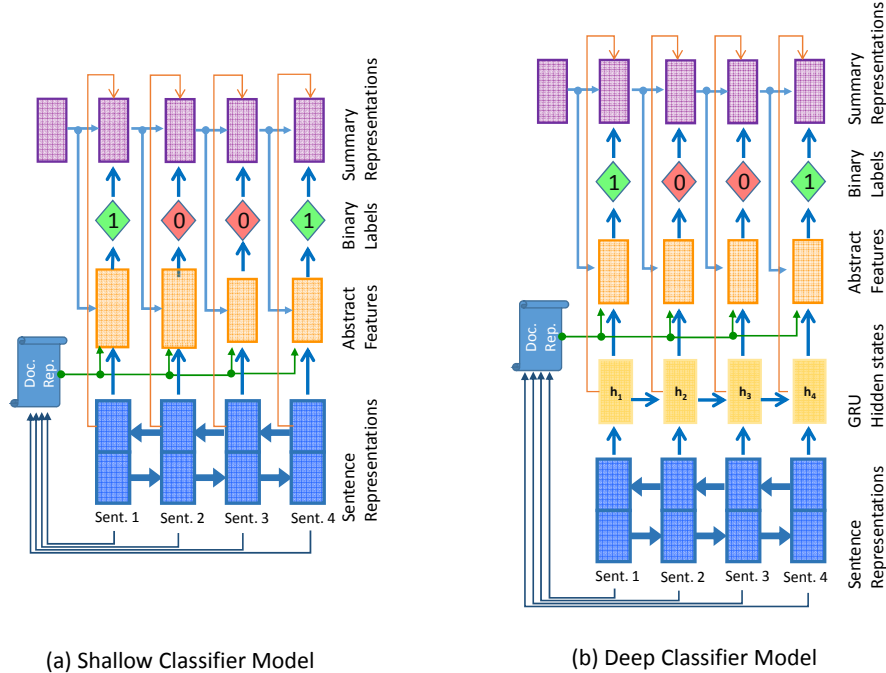

(a) Shallow Classifier Model

(b) Deep Classifier Model

Figure 1: The shallow and deep versions of the Classifier architecture for extractive summarization.

**Shallow Model**: In the shallow model, we estimate the dynamic summary representation as the running sum of the representations of the sentences visited so far weighted by their probability of being in the summary.

$$
\begin{aligned}
\mathbf{s}_j &= \sum_{i=1}^{j-1} \mathbf{h}_i y_i && \texttt{\#(training time)} \\
\mathbf{s}_j &= \sum_{i=1}^{j-1} \mathbf{h}_i P(y_i = 1|\mathbf{h}_i, \mathbf{s}_i, \mathbf{d}) && \texttt{\#(test time)}
\end{aligned}
$$

$$(3)$$

In other words, at training time, since the summary membership of sentences is known, the probabilities are binary, whereas at test time we use a weighted pooling based on the estimated probability

that each sentence belongs to the summary. There is no need to normalize the summary representations since the cosine similarity metric we use in the scoring function of Eq. (1) automatically normalizes them.

**Deep Model**: In the deep model, we introduce an additional layer of unidirectional sentence-level GRU-RNN that takes as input the sentence representations $\mathbf{h}_j$ at each time-step. The hidden state of the new GRU $\hat{\mathbf{h}}_j = GRU(\mathbf{h}_j)$ is used as a replacement for sentence representation $\mathbf{h}_j$ in computing summary membership scores using Eq. (1) as well as in computing the dynamic summary representation using Eq. (3). The main idea behind using this additional layer of GRU is to allow a greater degree of non-linearity in computing the summary representation.

The graphical representations of the shallow and deep models under the Classifier architecture are displayed in Figure 1 with their full set of dependencies.

## 2.2 SELECTOR ARCHITECTURE

In this architecture, the models do not make decisions in the sequence of sentence ordering; instead, they pick one sentence at a time in an order that they deem fit. The act of picking a sentence is cast as a sequential generative model in which one sentence-index is emitted at each time step that maximizes the score in Eq. 1. Accordingly, the probability of picking a sentence with index $I(j) = k \in \{1, \ldots, N_d\}$ at time-step $j$ is given by the `softmax` over the scoring function:

$$P(I(j) = k | \mathbf{s}_j, \mathbf{h}_k, \mathbf{d}) = \frac{\exp(\texttt{score}(\mathbf{h}_k, \mathbf{s}_j, \mathbf{d}, \mathbf{p}_k))}{\sum_{l \in \{1, \ldots, N_d\}} \exp(\texttt{score}(\mathbf{h}_l, \mathbf{s}_j, \mathbf{d}, \mathbf{p}_l))} \tag{4}$$

The loss function in this case is the negative log-likelihood of the selected sentences in the ground truth data as shown below.

$$\ell(\mathbf{W}, \mathbf{w}, \mathbf{b}) = -\sum_{d=1}^{N} \sum_{j=1}^{M_d} \log P(I(j)^{(d)} | \mathbf{h}_{I(j)^{(d)}}, \mathbf{s}_j^d, \mathbf{d}_d) \tag{5}$$

where $M_d$ is the number of sentences selected in the ground truth of document $d$, $\{I(1)^{(d)}, \ldots, I(M_d)^{(d)}\}$ is the ordered list of selected sentence indices in the ground truth of document $d$. The dependence of the loss function on the order of the selected sentences can be gauged by the fact that the probability of selecting a sentence at time step $j$ depends on the dynamic summary representation $\mathbf{s}_j$, which is estimated based on the all sentences selected up to time step $j-1$.

At test time, at each time-step, the model emits the index of the sentence that has the best score given the current summary representation as shown below.

$$I(j) = \arg \max_{k \in \{1, \ldots, N_d\}} \texttt{score}(\mathbf{h}_k, \mathbf{s}_j, \mathbf{d}, \mathbf{p}_k) \tag{6}$$

The estimation of dynamic summary representation is done differently for the shallow and deep selector models as described below.

**Shallow Model**: In this model, we sum the representations of the selected sentences until the time step $j$ as the dynamic summary representation. This is true for both training time and test time.

$$\mathbf{s}_j = \sum_{i=1}^{j-1} \mathbf{h}_{I(i)}. \tag{7}$$

**Deep Model**: In the deep model, we introduce an additional GRU-RNN whose time steps correspond to the sentence index emission events. At each time-step, it takes as input the representation of the previously selected sentence $\mathbf{h}_{I(j-1)}$, and computes a new hidden state $\hat{\mathbf{h}}_j = GRU(\mathbf{h}_{I(j-1)})$. Unlike the shallow model that maintains a separate vector for summary representation $\mathbf{s}_j$, we use $\hat{\mathbf{h}}_j$ as the summary representation $\mathbf{s}_j$ at time step $j$. This makes sense for the case of the Selector architecture since both at training and test time we make hard decisions of sentence selection, with the effect that the hidden state of the new GRU can capture a non-linear aggregation of the sentences selected until time step $j-1$.

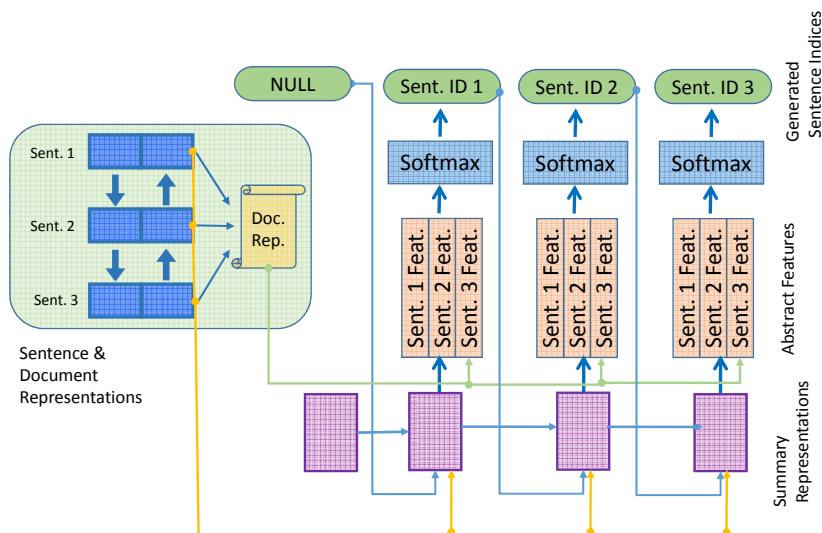

Figure 2: Selector architecture for extractive summarization. The shallow and deep versions are identical except for the fact that the simple vector representation for summary representation in the shallow version is replaced with a gated recurrent unit in the deep version.

Fig. 2 shows the graphical representation of the Selector architecture with all the dependencies between the nodes. The architecture is the same for both shallow and deep models with the only difference being that the simple summary representation in the former is replaced with a gated recurrent unit in the latter.

## 3 RELATED WORK

Previous researcher such as Shen et al. (2007) have proposed modeling extractive document summarization as a sequence classification problem using Conditional Random Fields. Our approach is different from theirs in the sense that we use RNNs in our model that do not require any handcrafted features for representing sentences and documents.

The Selector architecture broadly involves ranking of sentences by some criterion, therefore does correspond to traditional methods for extractive summarization such as TextRank (Mihalcea & Tarau (2004)) that also involve ranking of sentences by salience and novelty. However, to the best of our knowledge, our Selector framework is a novel deep learning framework for extractive summarization. Broader efforts are being made in the deep learning community to build more sophisticated sequence to sequence models towards the objective of automatically learning complex tasks such as sorting sequences (Oriol Vinyals (2015); Graves et al. (2014)), but their utility for extractive summarization remains to be explored.

In the deep learning framework, the extractive summarization work of Cheng & Lapata (2016) is the closest to our work. Their model is based on an encoder-decoder approach where the encoder learns the representation of sentences and documents while the decoder classifies each sentence using an attention mechanism. Broadly, their model is also in the Classifier framework, but architecturally, our approaches are different. While their approach can be termed as a multi-pass approach where both the encoder and decoder consume the same sentence representations, our approach is a deep one where the representations learned by the bidirectional GRU encoder are in turn consumed by the Classifier or Selector models. Another key difference between our work and theirs is that unlike our unsupervised greedy approach to convert abstractive summaries to extractive labels, Cheng & Lapata (2016) chose to train a separate supervised classifier using manually created labels on a subset of the data. This may yield more accurate gold extractive labels which may help boost the performance of their models, but incurs additional annotation costs.

## 4    EXPERIMENTS AND RESULTS

**Pseudo ground-truth generation**: In order to train our extractive Classifier and Selector models, for each document we need ground truth in the form of sentence-level binary labels and ordered list of selected sentences respectively. However, most summarization corpora only contain human written abstractive summaries as ground truth. To solve this problem, we use an unsupervised approach to convert the abstractive summaries to extractive labels. Our approach is based on the idea that the selected sentences from the document should be the ones that maximize the Rouge score with respect to gold abstractive summaries. Since it is computationally expensive to find a globally optimal subset of sentences that maximizes the Rouge score, we employ a greedy approach, where we add one sentence at a time incrementally to the summary, such that the Rouge score of the current set of selected sentences is maximized with respect to the entire gold summary. We stop adding sentences when either none of the remaining candidate sentences improves the Rouge score upon addition to the current summary set or when the maximum summary length is reached. We return this ordered list of sentences as the ground-truth for the Selector architecture. The ordered list is converted into binary summary-membership labels that are consumed by the Classifier architecture for training.

We note that similar approaches have been employed by other researchers such as Svore et al. (2007) to handle the problem of converting abstractive summaries to extractive ground truth. We would also like to point readers to a recent work by Cao et al. (2015) that proposes an ILP based approach to solve this problem optimally. Since this is not the focus of this work, we chose a simple greedy algorithm.

**Corpora**: For our experiments, we used the Daily Mail corpus originally constructed by Hermann et al. (2015) for the task of passage-based question answering, and re-purposed for the task of document summarization as proposed in Cheng & Lapata (2016) for extractive summarization and Nallapati et al. (2016a) for abstractive summarization. Overall, we have 196,557 training documents, 12,147 validation documents and 10,396 test documents from the Daily Mail corpus. On average, there are about 28 sentences per document in the training set, and an average of 3-4 sentences in the reference summaries. The average word count per document in the training set is 802.

We also used the DUC 2002 single-document summarization dataset[1] consisting of 567 documents as an additional out-of-domain test set to evaluate our models.

**Evaluation**: In our experiments below, we evaluate the performance of our models using different variants of the Rouge metric[2] computed with respect to the gold abstractive summaries. Following Cheng & Lapata (2016), we use limited length Rouge recall at 75 bytes of summary as well as 275 bytes on the Daily Mail corpus. On DUC 2002 corpus, following the official guidelines, we use limited length Rouge recall at 75 words. We report the scores from Rouge-1, Rouge-2 and Rouge-L, which are computed using matches of unigrams, bigrams and longest common subsequences respectively, with the ground truth summaries.

**Baselines**: On all datasets, we use *Lead-3* model, which simply produces the leading three sentences of the document as the summary, as a baseline. On the Daily Mail and DUC 2002 corpora, we also report performance of *LReg*, a feature-rich logistic classifier used as a baseline by Cheng & Lapata (2016). On DUC 2002 corpus, we report several baselines such as Integer Linear Programming based approach (Woodsend & Lapata (2010)), and graph based approaches such as TGRAPH (Parveen et al. (2015)) and URANK (Wan (2010)) which achieve very high performance on this corpus. In addition, we also compare with the state-of-the art deep learning supervised extractive model from Cheng & Lapata (2016).

**Experimental Settings**: We used 100-dimensional *word2vec* (Mikolov et al. (2013)) embeddings trained on the Daily Mail corpus as our embedding initialization. We limited the vocabulary size to 150K and the maximum sentence length to 50 words, to speed up computation. We fixed the model hidden state size at 200. We used a batch size of 32 at training time, and employed *adadelta* (Zeiler (2012)) to train our model. We employed gradient clipping and L-2 regularization to prevent overfitting and an early stopping criterion based on validation cost.

---

[1] http://www-nlpir.nist.gov/projects/duc/guidelines/2002.html
[2] http://www.berouge.com/Pages/default.aspx

At test time, for the Classifier models we pick sentences sorted by the predicted probabilities until we exceed the length limit as determined by the Rouge metric. Likewise, we allow the Selector models to emit sentence indices until the desired summary length is reached. For the Selector model, we also make sure the emitted sentence ids are not repeated across time steps by traversing down the sorted predicted probabilities of the `softmax` layer at each time step until we reach a sentence-id that was not emitted before.

We note that it is possible to optimize the Classifier performance at test time using the Viterbi algorithm to compute the best sequence of labels, subject to the Markovian assumptions of the architecture and model. Similarly, it is also possible to further boost the Selector's performance by using beam search at test time. However, in this work we used greedy classification/selection for inference since our primary interest is in comparing the two architectures, and our choice allows us to make a fair apples-to-apples comparison.

**Results on Daily Mail corpus**: Table 1 shows the performance comparison of our models with state-of-the-art model of Cheng & Lapata (2016) and other baselines on the DailyMail corpus using Rouge recall at two different summary lengths.

| Model | Recall at 75 bytes | | | Recall at 275 bytes | | |
|---|---|---|---|---|---|---|
| | Rouge-1 | Rouge-2 | Rouge-L | Rouge-1 | Rouge-2 | Rouge-L |
| Lead-3 | 21.9 | 7.2 | 11.6 | 40.5 | 14.9 | 32.6 |
| LReg(500) | 18.5 | 6.9 | 10.2 | N/A | N/A | N/A |
| Cheng '16 | 22.7 | 8.5 | 12.5 | **42.2** | **17.3**\* | 34.8 |
| Shal.-Select | 25.6 | 10.3 | 14.0 | 41.3 | 16.8 | 34.9 |
| Deep-Select | 26.1 | 10.7 | 14.4 | 41.3 | 15.3 | 33.5 |
| Shal.-Cls. | 26.0 | 10.5 | 14.23 | 42.1 | 16.8 | 34.8 |
| Deep-Cls. | **26.2**\* $\pm0.4$ | **10.7**\* $\pm0.4$ | **14.4**\* $\pm0.4$ | **42.2** $\pm0.2$ | 16.8 $\pm0.2$ | **35.0** $\pm0.2$ |

Table 1: Performance of various models on the **entire Daily Mail test set** using the **limited length recall** variants of Rouge with respect to the abstractive ground truth at **75 bytes** and **275 bytes**. Entries with asterisk are statistically significant using 95% confidence interval with respect to the nearest state-of-the-art model, as estimated by the Rouge script.

The results show that contrary to our initial expectation, the Classifier architecture is superior to the Selector architecture. Within each architecture, the deeper models are better performing than the shallower ones. Our deep classifier model outperforms Cheng & Lapata (2016) with a statistically significant margin at 75 bytes, while matching their model at 275 bytes. One potential reason our models do not consistently outperform the extractive model of Cheng & Lapata (2016) is the additional supervised training they used to create sentence-level extractive labels to train their model. Our models instead use an unsupervised greedy approximation to create extractive labels from abstractive summaries, and as a result, may generate noisier ground truth than theirs.

**Results on the Out-of-Domain DUC 2002 corpus**: We also evaluated the models trained on the DailyMail corpus on the out-of-domain DUC 2002 set as shown in Table 2. The performance trend is similar to that on Daily Mail. Our best model, Deep Classifier is again statistically on par with the model of Cheng & Lapata (2016). However, both models perform worse than graph-based TGRAPH (Parveen et al. (2015)) and URANK (Wan (2010)) algorithms, which are the state-of-the-art models on this corpus. Deep learning based supervised models such as ours and that of Cheng & Lapata (2016) perform very well on the domain they are trained on, but may suffer from domain adaptation issues when tested on a different corpus such as DUC 2002.

## 5 DISCUSSION

**Impact of Document Structure**: In all our experiments thus far, the classifier architecture has proven superior to the selector architecture. We conjecture that decision making in the same sequence as the original sentence ordering is perhaps advantageous in document summarization since there is a smooth sequential discourse structure in news stories starting with the main highlights of the story in the beginning, more elaborate description in the middle and ending with conclusive remarks. If this is true, then in scenarios where sentence ordering is less structured, the selector

|  | Rouge-1 | Rouge-2 | Rouge-L |
|---|---|---|---|
| Lead-3 | 43.6 | 21.0 | 40.2 |
| LReg | 43.8 | 20.7 | 40.3 |
| ILP | 45.4 | 21.3 | 42.8 |
| TGRAPH | 48.1 | **24.3**\* | - |
| URANK | **48.5**\* | 21.5 | - |
| Cheng *et al* '16 | 47.4 | 23.0 | **43.5** |
| Shallow-Selector | 44.6 | 20.0 | 41.1 |
| Deep-Selector | 45.9 | 21.5 | 42.4 |
| Shallow-Classifier | 45.9 | 21.5 | 42.3 |
| Deep-Classifier | 46.8 ±0.9 | 22.6 ±0.9 | 43.1 ±0.9 |

Table 2: Performance of various models on the **DUC 2002** set using the **limited length recall** variants of Rouge at **75 words**. Our Deep Classifier is statistically within the margin of error at 95% C.I. with respect to the model of Cheng & Lapata (2016), but both are lower than state-of-the-art results due to out-of-domain training.

architecture should be superior since it has freedom to select salient sentences in any arbitrary order. Such scenarios actually do occur in practice, *e.g.*, summarization of a cluster of tweets on a topic where there is no specific discourse structure between individual tweets, or in multi-document summarization where a pair of sentences across document boundaries have no specific ordering. In order to test this hypothesis, we simulated such data in the Daily Mail corpus by randomly shuffling the sentences in each document in the training set and retraining models under both the architectures, and evaluating them on the original test sets. The results, summarized in Table 3, show that the Classifier architecture suffers bigger losses than the Selector architecture when the document structure is destroyed. In fact, the Selector architecture performs slightly better than the Classifier architecture when trained on the shuffled data, indicating that our hypothesis may indeed be true.

|  | Trained on original data | | | Trained on shuffled sentences | | |
|---|---|---|---|---|---|---|
|  | Rouge-1 | Rouge-2 | Rouge-L | Rouge-1 | Rouge-2 | Rouge-L |
| Shallow-Selector | 41.3 | 16.8 | 34.9 | **40.6** | **15.6** | **33.0** |
| Shallow-Classifier | **42.1** | 16.8 | **35.0** | 40.1 | 15.3 | 32.9 |
| Deep-Selector | 41.3 | 15.3 | 33.5 | **40.5** | **15.3** | 32.5 |
| Deep-Classifier | **42.2** | **16.8** | **35.0** | 40.1 | 15.1 | **32.9** |

Table 3: Simulated experiment to demonstrate the impact of document discourse structure on model performance. Evaluation is done using Rouge limited length recall at 275 bytes. The Selector architecture exhibits superior performance when the discourse structure of the document is destroyed.

**Qualitative Analysis**: One of the advantages of our model design is teasing out various abstract features for the sake of interpretability of system predictions. In the appendix, we present a visualization (see Fig. 3 in the Appendix) of the system predictions based on the scores for various abstract features listed in Eq. (1). We also present the learned importance weights of these features in Table 4. A few representative documents are also presented in the appendix highlighting the sentences chosen by our models for summarization.

## 6 CONCLUSION AND FUTURE WORK

In this work, we propose two neural architectures for extractive summarization. Our proposed models under these architectures are not only very interpretable, but also achieve state-of-the-art performance on two different data sets. We also empirically compare our two frameworks and suggest conditions under which each of them can deliver optimal performance.

As part of our future work, we plan to further investigate the applicability of the novel Selector architecture to relatively less structured summarization problems such as summarization of multiple documents or topical clusters of tweets. In addition, we also intend to perform additional experiments on the Daily Mail dataset such as incorporating beam search in both model inference as well in pseudo ground truth generation that may result in further performance improvements.

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

# 7 APPENDIX

In this section, we will present some additional qualitative and quantitative analysis of our models that we hope will shed some light on their behavior.

## 7.1 VISUALIZATION OF MODEL OUTPUT

In addition to being state-of-the-art performers, our models have the additional advantage of being very interpretable. The clearly separated terms in the scoring function (see Eqn. 1) allow us to tease out various factors responsible for the classification/selection of each sentence. This is illustrated in Figure 3, where we display a representative document from our validation set along with normalized scores from each abstract feature from the deep classifier model. Such visualization is especially useful in explaining to the end-user the decisions made by the system.

## 7.2 LEARNED IMPORTANCE WEIGHTS

We display in Table 4 the learned importance weights corresponding to various abstract features for deep sentence selector. Confirming our intuition, the model learns that salience and redundancy are the most important predictive features for summary membership of a sentence, followed by positional feature and content based feature. Further, when the same model is trained on documents with randomly shuffled sentences, it learns very small weight for the positional features, which is exactly what one expects.

| Training condition | Salience | Content | Position | Redundancy |
|---|---|---|---|---|
| Original data | 42.75 | 14.83 | -31.09 | 40.99 |
| Shuffled data | 9.69 | 2.85 | 0.20 | 16.08 |

Table 4: Learned weights of various abstract features from the deep sentence selector model. Salience and redundancy are the most important features as learned by the model, followed by position and content. The negative sign for position weights has no particular significance. The positional feature gets very low weight when the document structure is destroyed by randomly shuffling sentences in each document the training data.

| Gold Summary:<br>Redpath has ended his eight-year association with Sale Sharks. Redpath spent five years as a player and three as a coach at sale. He has thanked the owners, coaches and players for their support. | Salience | Content | Novelty | Position | Prob. |
|---|---|---|---|---|---|
| Bryan Redpath has left his coaching role at Sale Sharks with immediate effect. | 0.1 | 0.1 | 0.9 | 0.1 | 0.3 |
| The 43 - year - old Scot ends an eight-year association with the Aviva Premiership side, having spent five years with them as a player and three as a coach. | 0.9 | 0.6 | 0.9 | 0.9 | 0.7 |
| Redpath returned to Sale in June 2012 as director of rugby after starting a coaching career at Gloucester and progressing to the top job at Kingsholm . | 0.8 | 0.5 | 0.5 | 0.9 | 0.6 |
| Redpath spent five years with Sale Sharks as a player and a further three as a coach but with Sale Sharks struggling four months into Redpath's tenure, he was removed from the director of rugby role at the Salford-based side and has since been operating as head coach . | 0.8 | 0.9 | 0.7 | 0.8 | **0.9** |
| 'I would like to thank the owners, coaches, players and staff for all their help and support since I returned to the club in 2012. | 0.4 | 0.1 | 0.1 | 0.7 | 0.2 |
| Also to the supporters who have been great with me both as a player and as a coach,' Redpath said. | 0.6 | 0.0 | 0.2 | 0.3 | 0.2 |

Figure 3: Visualization of Deep Classifier output on a representative document. Each row is a sentence in the document, while the shading-color intensity in the first column is proportional to its probability of being in the summary, as estimated by the scoring function. In the columns are the normalized scores from each of the abstract features in Eqn. (1) as well as the final prediction probability (last column). Sentence 2 is estimated to be the most salient, while the longest one, sentence 4, is considered the most content-rich, and not surprisingly, the first sentence the most novel. The third sentence gets the best position based score.

## 7.3 ABLATION EXPERIMENTS

We evaluated the performance of the deep selector and deep classifier models on the validation set by deleting one abstract feature at a time from the model, with replacement. The performance numbers, displayed in Table 5, show that removing any of the features results in a small loss in performance. Note that the priority of features in the ablation experiments need not correspond to their priority in terms of learned weights in Table 4, since feature correlations may affect the two metrics differently. For the deep classifier, content and redundancy seem to matter the most while for the deep selector, dropping positional features hurts the most. Based on this analysis, we plan to investigate more thoroughly the reasons behind the poor ablation performance of salience and redundancy in the classifier and selector models respectively.

| Features | Deep Classifier | | | Deep Selector | | |
|---|---|---|---|---|---|---|
| | Rouge-1 | Rouge-2 | Rouge-L | Rouge-1 | Rouge-2 | Rouge-L |
| All | 42.43 | 17.32 | 34.07 | 41.55 | 16.52 | 32.41 |
| -Salience | 42.40 | 17.27 | 34.09 | 40.82 | 15.99 | 31.45 |
| -Position | 41.78 | 16.76 | 33.58 | **39.06** | **14.32** | **29.85** |
| -Content | **41.12** | **15.78** | 33.23 | 40.68 | 15.83 | 31.13 |
| -Redundancy | 41.67 | 16.86 | **32.93** | 41.46 | 16.50 | 32.31 |

Table 5: Ablation experiments on the validation set to gauge the relative importance of each abstract feature. The top row is where all four abstract features are present. The following rows corresponding to removal of one feature at a time with replacement. Evaluation is done using Rouge limited length recall at 275 bytes. Bold faced entries correspond to largest reduction in performance when the corresponding features are dropped.

## 7.4 REPRESENTATIVE DOCUMENTS AND EXTRACTIVE SUMMARIES

We display a couple of representative documents, one each from the Daily Mail and DUC corpora, highlighting the sentences chosen by deep classifier and comparing them with the gold summaries in Table 6. The examples demonstrate qualitatively that the model performs a reasonably good job in identifying the key messages of a document.

*Document:* **@entity0 have an interest in @entity3 defender @entity2 but are unlikely to make a move until january** . **the 00 - year - old @entity6 captain has yet to open talks over a new contract at @entity3 and his current deal runs out in 0000** . @entity3 defender @entity2 could be targeted by @entity0 in the january transfer window @entity0 like @entity2 but do n't expect @entity3 to sell yet they know he will be free to talk to foreign clubs from january . @entity12 will make a 0million offer for @entity3 goalkeeper @entity14 this summer . the 00 - year - old is poised to leave @entity16 and wants to play for a @entity18 contender . **@entity12 are set to make a 0million bid for @entity2 's @entity3 team - mate @entity14 in the summer**

*Gold Summary:* @entity2 's contract at @entity3 expires at the end of next season . 00 - year - old has yet to open talks over a new deal at @entity16 . @entity14 is poised to leave @entity3 at the end of the season

---

*Document:* **today , the foreign ministry said that control operations carried out by the corvette spiro against a korean-flagged as received ship fishing illegally in argentine waters were carried out " in accordance with international law and in coordination with the foreign ministry " .** the foreign ministry thus approved the intervention by the argentine corvette when it discovered the korean ship chin yuan hsing violating argentine jurisdictional waters on 00 may . ... **the korean ship , which had been fishing illegally in argentine waters , was sunk by its own crew after failing to answer to the argentine ship 's warnings .** the crew was transferred to the chin chuan hsing , which was sailing nearby and approached to rescue the crew of the sinking ship .....

*Gold Summary:* the korean-flagged fishing vessel chin yuan hsing was scuttled in waters off argentina on 00 may 0000 . adverse weather conditions prevailed when the argentine corvette spiro spotted the korean ship fishing illegally in restricted argentine waters . the korean vessel did not respond to the corvette 's warning . instead , the korean crew sank their ship , and transferred to another korean ship sailing nearby . in accordance with a uk-argentine agreement , the argentine navy turned the surveillance of the second korean vessel over to the british when it approached within 00 nautical miles of the malvinas ( falkland ) islands .

Table 6: Example documents and gold summaries from Daily Mail (top) and DUC 2002 (bottom) corpora. The sentences chosen by deep classifier for extractive summarization are highlighted in bold.

