# Peer review of "Classify or Select: Neural Architectures for Extractive Document Summarization"

_ICLR 2017 — rejected_

[Public Comment · Gaurav Bhaskar Gite · 15 Nov 2016]
**Clarifying question on positional embedding (Pj) of the sentence?**

Very interesting paper.
I have one clarifying question.
How are the embeddings for the forward and backward position indices of the sentence in the document computed? Basically, I want to understand how the positional embedding for the sentences (Pj) calculated. 
Thank you.

[Official Review · AnonReviewer2 · rating 4 · confidence 4 · 17 Dec 2016]
**feedback**

This paper presents two RNN architectures for extractive document summarization. The first one, Classifier, takes into account the order in which sentences appear in the original document, whereas the second one, Selector, chooses sentences in an arbitrary order. For each architecture, the concatenated RNN hidden state from a sentence forward and backward pass  is used as features to compute a score that captures content richness, salience, positional importance, and redundancy. Both models are trained in a supervised manner, so the authors used "pseudo-ground truth generation" to create training data from abstractive summaries. Experiments show that the Classifier model performs better, and it achieves near state-of-the-art performance for some evaluation metrics.

The proposed model is in general an extension of Cheng and Lapata, 2016. Unfortunately, the performance is only slightly better or sometimes even worse. The authors mentioned that one key difference how they transform abstractive summaries to become gold labels for the supervised method. However, in the experiment results, the authors described that one potential reason their models do not consistently outperform the extractive model of Cheng & Lapata, 2016 is that the unsupervised greedy approximation may generate noisier ground truth labels than Cheng & Lapata. Is there a reason to construct the training data similar to Cheng & Lapata, if that turns out to be a better method?
In order for the proposed models to be convincing, they need to outperform this baseline that's very similar to the proposed methods more consistently, since the main contribution is improved neural architectures for extractive document summarization.

[Official Review · AnonReviewer1 · rating 4 · confidence 4 · 17 Dec 2016]
**A RNN model for extractive summarization**

This paper provides two RNN-based architectures for extractive document summarization. The first, "Classify", reads in the whole document and traverses the sentences a second time to decide whether to include them or not (0/1 decisions). The second, "Select",  reads in the whole document and picks the most relevant sentence one at the time. The models assume that oracle extractive summaries exist, and a pseudo ground-truth generation procedure is used, which mimics Svore et al. (2007) among others. 

Overall, this paper seems a small increment over Cheng & Lapata (2016) and performance is similar or worse to that paper. The problem of single document extractive summarization is not particularly exciting since in DUC 2002 (14 years ago) existing models could not beat the lead baseline (which selects the first sentences of the document). It's a pity that this paper doesn't address the most interesting problems of abstractive summarization or apply the proposed approach to multi-document summarization. It's also a little disappointing that the maximum sentence length had to be capped to 50, which suggests the model has some trouble to scale.

[Official Review · AnonReviewer3 · rating 6 · confidence 4 · 19 Dec 2016]
**Interesting models**

This paper presents two models for extractive document summarization: the classifier architecture and the selector architecture. These two models basically use either classification or ranking in a sequential order to pick the candidate sentences for summarization. Experiments in this paper show the results are either better or close to the SOTA.

Technical comments:

- In equation (1), there is a position-relevant component call "positional importance". I am wondering how important this component is? Is it possible to show the performance without this component? Especially, for the discussion on impact of document structure, when the model is trained on the shuffled order but tested on the original order.
- A similar question about equation (1), is the content-richness component really necessary? Since the score function already has salience part, which could measure how important of $h_j$ with respect to the whole document.
- For the dynamic summary representation in equation (3), why not use the same updating equation for both training and test procedures? During test time, the model actually knows the decisions that have been made so far by the decoder. In this way, the model will be more consistent during training and test. 
- I think section 5 is the most interesting part of this paper, and it is also convincing on the difference between the two architectures.
- It is a little disappointing that the decoding algorithm used in this paper is too simple. In a minimal case, both of them could use beam search and the results could be better.

[Final Decision · Program Chairs · 06 Feb 2017]
**ICLR committee final decision**

Reviewers found this paper clear to read, but leaned negative on in terms of impact and originality of the work. Main complaint is that the paper is neither significantly novel in terms of modeling (pointing to Cheng & Lapata), nor significantly more performative on this task ("only slightly better"). One reviewer also has a side complaint that the task itself is also somewhat simplistic and simplified, and suggests other tasks. This comment is perhaps harsh, but reflects a mandate for revisiting "old" problems to provide significant improvements in accuracy or novel modeling.